# The range and reach of qualitative research in neurosurgery: A scoping review

Charlotte J. Whiffin[1,2,3]*, Kathleen Joy O. Khu[4], Brandon G. Smith[2,3], Isla Kuhn[5], Santhani M. Selveindran[6], Laura Hobbs[2,3,7], Samin Davoody[8], Yusuf Docrat[9], Orla Mantle[2,3], Upamanyu Nath[10], Lara Onbaşı[11], Stasa Tumpa[12], Ignatius N. Esene[13], Harry Mee[12], Fergus Gracey[14], Shobhana Nagraj[15], Tom Bashford[2,3,16], Angelos G. Kolias[2], Peter J. Hutchinson[2]

1 College of Health, Psychology and Social Care, University of Derby, Derby, United Kingdom, 2 NIHR Global Health Research Group on Acquired Brain and Spine Injuries, University of Cambridge, Cambridge United Kingdom, 3 International Health Systems Group, Department of Engineering, University of Cambridge, 4 Division of Neurosurgery, Department of Neurosciences, College of Medicine and Philippine General Hospital, University of the Philippines Manila, Philippines, 5 Medical Library, University of Cambridge, United Kingdom, 6 Open Health, Marlow, United Kingdom, 7 East and North Hertfordshire NHS Trust, Stevenage, United Kingdom, 8 Student Research Committee, School of Medicine, Shahid Beheshti University of Medical Sciences, Tehran, Iran, 9 Division of Neurosurgery, Faculty of Medicine and Health Sciences, Stellenbosch University, Cape Town South Africa, 10 Wythenshawe Hospital, Manchester University NHS foundation Trust, Manchester, United Kingdom, 11 Hacettepe University, Faculty of Medicine & Hospital Ankara Turkey, 12 Addenbrooke's Hospital, Cambridge University Hospitals NHS Foundation Trust, Cambridge United Kingdom, 13 Neurosurgery Division, Faculty of Health Sciences, University of Bamenda, Cameroon, 14 Department of Clinical Psychology and Psychological Therapies, University of East Anglia, Norwich, United Kingdom, 15 Department of Public Health & Primary Care, University of Cambridge, 16 Division of Anaesthesia, University of Cambridge, Addenbrooke's Hospital, Cambridge, United Kingdom

* cw753@medschl.cam.ac.uk

## Abstract

Following calls for more qualitative research in neurosurgery, this scoping review aimed to describe the range and reach of qualitative studies relevant to the field of neurosurgery and the patients and families affected by neurosurgical conditions. A systematic search was conducted in September 2024 across six databases: Medline via Ebsco; Embase via OVID; PsycINFO via Ebsco; Scopus; Web of Science Core Collection; and Global Health via Ebsco. Eligibility criteria were based on Population, Concept, and Context. The search identified 18,809 hits for screening with 812 included in the final analysis. Seven themes were identified from a content analysis of study aims: 1 Perspectives of living with a neurosurgical condition; 2 Family perspectives; 3 Perceptions of neurosurgery; 4 Perceptions of general healthcare care; 5 Decision making; 6 Advancing neurosurgery; and, 7 Understanding neurosurgical conditions. Traumatology was identified as the most researched sub-specialty (43.2%) yet few studies were led explicitly by a neurosurgeon (1.6%) or those with a neurosurgical affiliation (10.5%). Lead authors were predominantly from high income countries (93.7%), as were most multi-author teams (86.6%). There was a trend towards increasing publication over time; however, only 8.4% of papers

**Data availability statement:** All relevant data are within the manuscript and its Supporting information files.

**Funding:** This research was funded by the NIHR (NIHR132455) using UK international development funding from the UK Government to support global health research. The views expressed in this publication are those of the author(s) and not necessarily those of the NIHR or the UK government."

**Competing interests:** The authors have declared that no competing interests exist.

were published in neurosurgical specific journals. The data set had an average Field Weighted Citation Impact of 0.96 and Field Weighted Views Impact of 1.11, 18.9% were cited in policy documents in 15 countries. This scoping review provides a comprehensive picture of the current qualitative research base in neurosurgery and suggests ways to improve the conduct and reporting of such studies in the future. Addressing these challenges is crucial if qualitative research is to advance the neurosurgical evidence base in a rigorous way.

## Introduction

In the context of neurosurgery, where objectivity and quantitative outcomes traditionally dominate the evidence base, the role of qualitative research has often been overlooked [1]. Yet, qualitative research plays a critical role in healthcare, offering profound insights into the complex interplay of clinical practices, patient journeys, and familial experiences that objective outcomes struggle to capture [2,3].

In recent years, there has been a growing recognition of the value that qualitative methodologies bring to surgical specialties [1,4–6], particularly in trials. Such studies provide important insights into the social context in which trials are conducted explaining what worked well and what failed to work as expected [7–9]. The recent Lancet Neurology Commission on Traumatic Brain Injury (TBI) also demonstrated the value of qualitative insights to understand the complex challenges of collecting outcome data in Low- and Middle-Income Countries (LMICs) [10,11] and repeated calls for more recognition by peers and policy makers of the value of this methodology [1,10].

Despite this, the scientific paradigm of modern medicine prioritises objectivity over subjectivity and deduction over induction leaving qualitative research omitted from traditional evidence hierarchies or subsumed into the lowest category of evidence alongside expert opinion and anecdotal findings [2,12,13]. While there is a steady growth of qualitative research published in general medical journals [12], it is still discounted by many as an inferior research design of little value to practice, and a low priority for publication [3]. This marginalization is particularly apparent in surgical disciplines and despite calls for the integration of qualitative findings into the evidence base, very little is published in high impact journals [13] with this bias attributed to journals' preference for generalisable results [14].

While there have been attempts to quantify the contribution of qualitative research to general [14] and rural surgery [15] there has yet to be such quantification of qualitative research in neurosurgery. Therefore, the aim of this scoping review was to determine the range and reach of qualitative studies relevant to the field of neurosurgery and the patients and families affected by neurosurgical conditions.

## Materials and methods

Given the primary aim of this review was to determine the range and reach of the qualitative evidence base, a review methodology that would determine its nature and volume [16] was deemed more appropriate than a systematic review or qualitative evidence synthesis.

Arksey and O'Malley's [16] review methodology was followed, with minor refinements as suggested by Levac et al. [17] and the Joanna Briggs Institute (JBI) [18].

The purpose of this scoping review was fourfold:

1. To examine the extent, range, and nature of research activity.

2. To determine the value for undertaking a full systematic review.

3. To summarize and disseminate research findings.

4. To identify research gaps in the existing literature.

An iterative approach was used "requiring researchers to engage with each stage in a reflexive way and, where necessary, repeat steps to ensure that the literature is covered in a comprehensive way" [16]. Due to these reflexive stages the protocol was not made publicly available. Reporting of this scoping review is in accordance with the Preferred Reporting Items for Systematic Reviews and Meta-Analyses extension for Scoping Reviews (PRISMA-ScR) (see S1 Table. PRISMA-ScR completed checklist) [19].

## Stage 1: Identifying the research question

Based on the Population, Concept and Context model (PCC) [18] the scoping review question was: *What qualitative studies have investigated neurosurgical conditions and treatments from the perspective of neurosurgeons, patients and families?* The population in question was defined as neurosurgeons (all training grades), patients, and informal carers/family members. The context was defined as a neurosurgical condition and/or treatment or a neurological condition where surgery was involved (e.g., surgical interventions for epilepsy). The concept of interest was 'qualitative research', defined as "a research study that uses a qualitative method of data collection and analysis" [20].

## Stage 2: Identifying relevant studies

An expert subject librarian (IK) developed a comprehensive search strategy deployed in six databases from inception to September 2024: Medline via Ebsco; Embase via OVID; PsycINFO via Ebsco; Scopus; Web of Science Core Collection; and Global Health via Ebsco (see S2 Table. Full Medline search strategy). Validated search filters for 'qualitative research' were adopted and a bespoke search for neurosurgery or neurosurgical conditions developed using the list of neurosurgical conditions and treatments from the American Association of Neurological Surgeons [21]. Validity of the search was confirmed against a core set of papers [22–33].

## Stage 3: Study selection

The search yielded 18,809 hits for screening. Eligibility criteria were based on PCC (see Table 1), in addition to a language restriction of publications written in English. Studies recruiting members of the public were included where topics were relevant to a neurosurgical condition, e.g., prevention of TBI. Samples of healthcare professionals without reference to the inclusion of neurosurgeons specifically were excluded. Qualitative evaluations of interventions relevant to the acute/sub-acute period were included, those focused on rehabilitation interventions excluded. Mixed methods were included where they met the definition of qualitative research. Only peer reviewed published primary research was included, and grey literature was not reviewed.

Screening of titles and abstracts was conducted using Rayyan software [34]. First, a test set of 50 papers were identified and in/out decisions discussed. Pairs then double-blind screened titles with review by abstract conducted by KK (an experienced neurosurgeon) and CW (a registered nurse and experienced qualitative researcher). In total n = 1,178 were assessed by full text for eligibility.

**Table 1. Inclusion/Exclusion criteria.**

| | | Inclusion | Exclusion |
|---|---|---|---|
| **Population** | Neurosurgeons, patients, carers, trainees | Studies that recruit neurosurgeons or trainees, neurosurgical patients and or their carers, peers/ public. | Nurses, midwives, and allied health professionals/trainees, teachers. |
| **Context** | Neurosurgical treatments and/or conditions | Investigates a topic relevant to a neurosurgical condition and/or treatment.<br>Neurological condition where surgery is involved.<br>Qualitative evaluations of interventions within the acute/sub-acute phase. | Outside the scope of 'neurosurgery'.<br>e.g., cluster headaches but no surgery implicated in population; epilepsy without surgical interventions mentioned; encephalitis, head and neck, stroke unless sub arachnoid haemorrhage mentioned or surgery mentioned.<br>Qualitative evaluations of interventions within the chronic/rehabilitation phase, e.g., goal setting/tele-rehab/rehab service delivery. |
| **Concept** | Qualitative research | Reported as a qualitative study, collects qualitative data AND conducts a qualitative analysis.<br>Peer reviewed published primary research.<br>Purist and pluralist studies (i.e., mixed methods). | Reports non-numerical data without using qualitative analysis. Quantitative research, Secondary research, commentaries, discussion papers, editorials, conference proceedings/abstracts, theses/ dissertations. |

## Stage 4: Charting the data

Each pair extracted and then verified information reported in the published manuscript. Data fields were identified *a priori* and data validation used in a Microsoft Excel spreadsheet to reduce variability. Where *a priori* categories were insufficient bespoke extraction was allowed.

## Stage 5: Collating, summarizing, and reporting results

Data aggregation provided summary data for most of the data extraction fields using descriptive statistics. A conventional content analysis [35] was then used to interrogate the aims of each manuscript. This approach avoids preconceived categories allowing these to be drawn directly from the data. Exact words from the aims were coded to capture core concepts. Codes were then sorted into categories and overarching themes. The coding of aims was conducted by CW using a qualitative analysis software program, NVivo 14. Further to this, data were used to generate word clouds. Word clouds are visualizations of word frequencies where words that occur more frequently are larger, therefore indicative of patterns and areas of commonality. Finally, we used SciVal.com to conduct a bibliometric analysis of the data set to calculate the average Field Weight Citation Impact (FWCI) and average Field-Weighted Views Impact (FWVI) (where a value of 1 means that the output performs as expected against the global average) and the citation of outputs in policy documents [36]. Once familiar with the evidence base, and its complexities, the results were combined under five themes: the what, the who, the how, the where, and the use of qualitative research in neurosurgery.

## Stage 6: Consultation

A scoping review framework can include a final stage of consultation. In this review we engaged several experts from the fields of neurosurgery and qualitative research. For this step we asked for their insights and perspectives on the findings of the review, the interpretation of this analysis and the subsequent recommendations.

## Results

There were 812 papers included in this review (see Fig 1 [37] and S3 Table. data extraction). Given the size of the evidence base our findings only present summary themes. Specific studies are discussed later as exemplars of important issues.

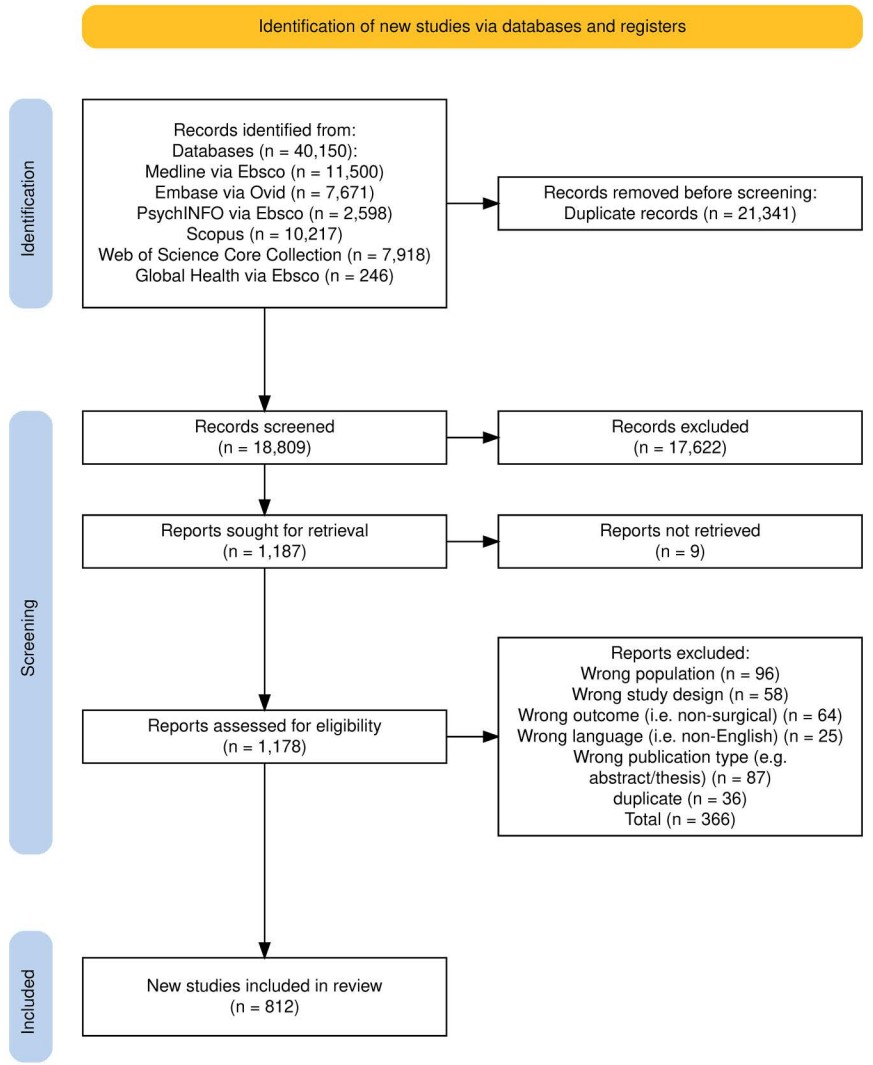

**Fig 1. PRISMA.** Flow diagram of literature acquisition leading to final selection of papers for the scoping review.

## The 'what'

This first theme presents the range of topics examined through qualitative methodologies. Central to this theme was understanding the study aims, whose perspective the researchers were investigating, and how well each neurosurgical subspecialty was represented. Seven themes were identified from the content analysis of aims and listed here in order of coverage: 1 Perspectives of living with a neurosurgical condition; 2 Family perspectives; 3 Perceptions of neurosurgery; 4 Perceptions of general healthcare care; 5 Decision Making; 6 Advancing neurosurgery; 7 Understanding neurosurgical conditions. Associated sub-themes are summarised in Table 2 and Fig 2.

Participants were most often patients (n = 283, 34.9%), or family members (n = 183, 22.5%), or a mix of patients and family members (n = 128, 15.8%). Neurosurgeons were the primary subject in n = 31 studies (3.8%) or part of a mixed group of healthcare professionals in n = 54 studies (6.7%). Participants were mostly recruited from HICs (n = 714, 88.0%), with only n = 54 (6.7%) recruited in LMICs and n = 5 (0.6%) from both.

**Table 2. Themes and sub-themes.**

| Themes and sub-themes | Count |
|---|---|
| **1 Perspectives of living with a neurosurgical condition** | **270** |
| 1.1 Impact study, e.g., lived experiences of having a neurosurgical condition inc. spirituality, life goals, Post Traumatic Growth, meaning, culture, gender, life satisfaction, adjustment, adaptation, identity, self, change. | 105 |
| 1.2 Symptoms, outcomes, concerns and difficulties, e.g., Mental health, sexuality, migraines, hypersensitivity, speech, fatigue, pain. | 56 |
| 1.3 Return to productivity, e.g., work, school, education, driving. | 38 |
| 1.4 Quality of Life and wellbeing. | 30 |
| 1.5 Familial and Social Capital, e.g., relationships, friendship, support, participation. community reintegration, inclusion. | 29 |
| 1.6 Needs, e.g., support needs, spiritual needs, educational needs. | 12 |
| **2 Family perspectives** | **194** |
| 2.1 Impact study, e.g., lived experiences of being a family member, stress, coping, anxiety, resilience, meaning, sense making, uncertainty, ambiguous loss, grief, change, adjustment, adaptation. | 86 |
| 2.2 Roles and relationships, familial, marital, sibling, sexual, spouses, parents, siblings. | 46 |
| 2.3 Providing care and support, managing challenging behaviours, communication difficulties, vigilance. | 31 |
| 2.4 Needs, e.g., support needs, general needs, support networks. | 25 |
| 2.5 Quality of life and wellbeing. | 6 |
| **3 Perceptions of neurosurgery** | **173** |
| 3.1 Patient experiences of neurosurgery, post operative recovery and rehabilitation. | 97 |
| 3.2 Acceptability and feasibility of neurosurgical procedures. | 34 |
| 3.3 Attitudes, understanding and perceptions of neurosurgery (Lay and clinician) | 31 |
| 3.4 Preferences and expectations of neurosurgery. | 11 |
| **4 Perceptions of general healthcare care** | **124** |
| 4.1 Experiences of, and satisfaction with, healthcare provision, delivery, pathways and systems. | 65 |
| 4.2 Transitions in care, e.g., between acute care settings, acute care to rehabilitation, hospital to home, | 32 |
| 4.3 Provider communication, e.g., communication styles, barriers to, preferences for. | 16 |
| 4. Palliative and support care needs. | 11 |
| **5 Decision making** | **81** |
| 5.1 Patient and family treatment decisions, e.g., shared decision making, hope, optimism, barriers and facilitators, uncertainty, information needs. | 44 |
| 5.2 Clinical decision making, e.g., management, surgical intervention, prognostic uncertainty, treatment options. | 38 |
| **6 Advancing neurosurgery** | **72** |
| 6.1 Research endeavour, e.g., Instrument development, Core sets PROMS, scale validation, Capacity building and patient engagement in research. | 23 |
| 6.2 Professional insights. e.g., gender, identity, retention, consensus, surgeon patient relationships. | 23 |
| 6.3 Neurosurgical innovation, e.g., Novel technology, equipment donation, AI | 15 |
| 6.4 Neurosurgical education and training (Experiences of…) | 11 |
| **7 Understanding neurosurgical conditions** | **51** |
| 7.1 Patient education and Information, e.g., needs, sources, health literacy, interventions. | 24 |
| 7.2 Social media use and content. | 14 |
| 7.3 Public health and injury antecedents, e.g., Risk perception, helmet use, concussion reporting, knowledge and beliefs, infant crying. | 13 |

Analysis of papers by Royal College of Surgeons neurosurgical subspecialty [38] showed traumatology had the largest qualitative evidence base (n = 351, 43.2%), followed by functional neurosurgery (n = 92, 11.3%), spine surgery (n = 79, 9.7%), neuro-oncology (n = 64, 7.9%), paediatric (n = 32, 3.9%), neurovascular (n = 26, 3.2%) and skull base (n = 7, 0.9%) (Fig 3). A word cloud generated from data extracted about more specific conditions/ pathologies/ procedures showed TBI had the highest frequency count followed by mixed Acquired Brain Injury (ABI), spinal procedures, brain tumours and deep brain stimulation (Fig 4).

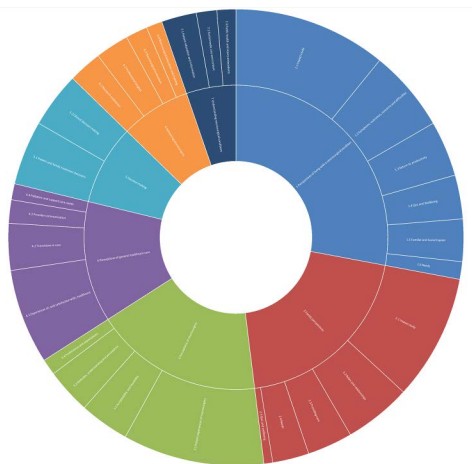

**Fig 2. Sunburst chart of themes and sub-themes.** This sunburst diagram presents the coverage of themes and sub-themes across the evidence base.

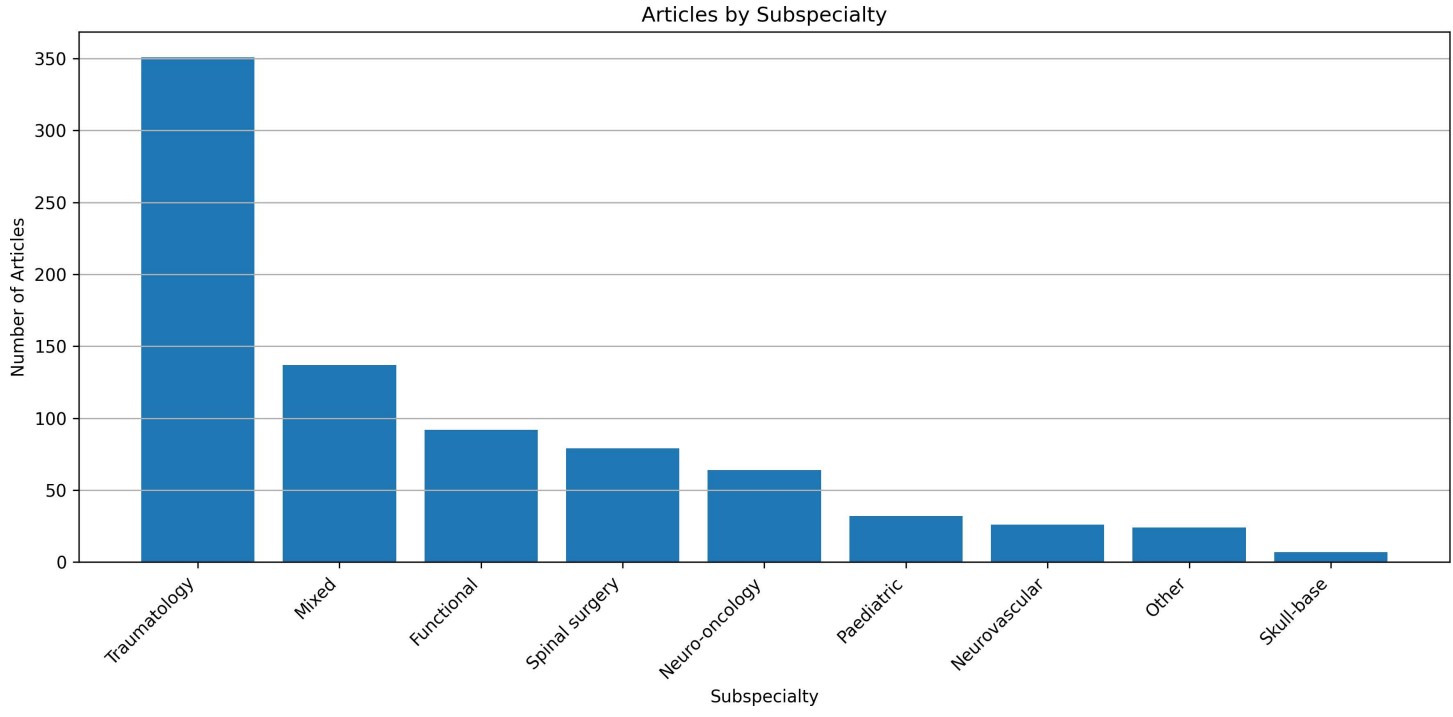

**Fig 3. Articles by subspecialty.** Bar chart displaying the number of papers in the review separated by neurosurgical subspecialty.

Almost two thirds (n = 504, 62.1%) related to adult neurosurgical conditions, 13.7% (n = 111) to paediatrics and 4.8% (n = 39) were specific to adolescent experiences.

### The 'who'

The second theme considered who was conducting the research. Data pertaining to profession and affiliation were examined in addition to what country the lead author originated and if they were working in HIC/LMIC multi-author teams.

**Fig 4. Word cloud of treatments and conditions.** Visual representation of word frequencies associated with neurosurgical treatments and conditions (visualisation performed using NVivo 14).

Most authors n = 466 (57.5%) did not report their profession and of those that did, the most frequent professional groups were nurses (n = 91, 11.2%) followed by physicians/MDs (n = 58, 7.2%) with only 13 authors (1.6%) explicitly identifying themselves as a neurosurgeon or resident. A text-based analysis was conducted to locate lead authors with a neurosurgical affiliation and found this explicitly stated in only n = 85 (10.5%).

Analysis of lead author by country, showed n = 761 (93.7%) were from HICs [39] (top three: n = 234 United States of America, n = 141 Canada, n = 107 UK) compared to only n = 47 (5.8%) from LMICs (Fig 5) (top three: Iran n = 14, South Africa n = 9, Thailand n = 6). Most multi-author teams were all from HICs (n = 703, 86.6%); with only n = 41 (5.1%) all from LMICs, and n = 32 (3.9%) with mixed HIC/LMIC authorship.

Only n = 10 manuscripts were identified where the lead author was from a HIC, but data were collected in a single LMIC setting and n = 4 studies where data collection was in an LMIC, and the study authored by an all HIC authorship team.

### The 'how'

The third theme examined the methods used and included the research methodology/research designs as stated, the reporting of paradigms/theoretical frameworks, sample sizes, data collection techniques, use of analytical theory, and reporting guidelines. Almost half of the studies (n = 461, 56.8%) described their study as qualitative, or descriptive qualitative, and did not specify further. Of those that described their study as a specific research design, n = 69 (8.5%) used a variant of phenomenology (including descriptive, interpretive and interpretive phenomenological analysis), n = 55 (6.8%) used grounded theory, n = 25 (3.1%) were described as a case study, n = 10 (1.2%) ethnography, and n = 8, (1.0%) narrative. There was n = 24 (3.0%) surveys and n = 115 (14.2%) mixed or multiple methods.

A descriptive analysis on the use of paradigms/theory was problematic given the lack of consistency in reporting. Where it was possible to identify some relevant framework or underpinning philosophy, this was only evident in n = 239 (29.4%).

In single method studies, sample sizes ranged from 1–173 for interview methods and 3–150 for focus group studies (Fig 6.) with larger samples associated with mixed method/qualitative surveys and documentary sources. Analysis of social media sources could also facilitate larger samples (e.g., analysis of 'Tweets', online forums and YouTube videos).

Data collection methods included semi-structured/in-depth interviews, focus groups, observations, essays, open questions in surveys, social media content and videos. Less common, but still present, were creative methods such as

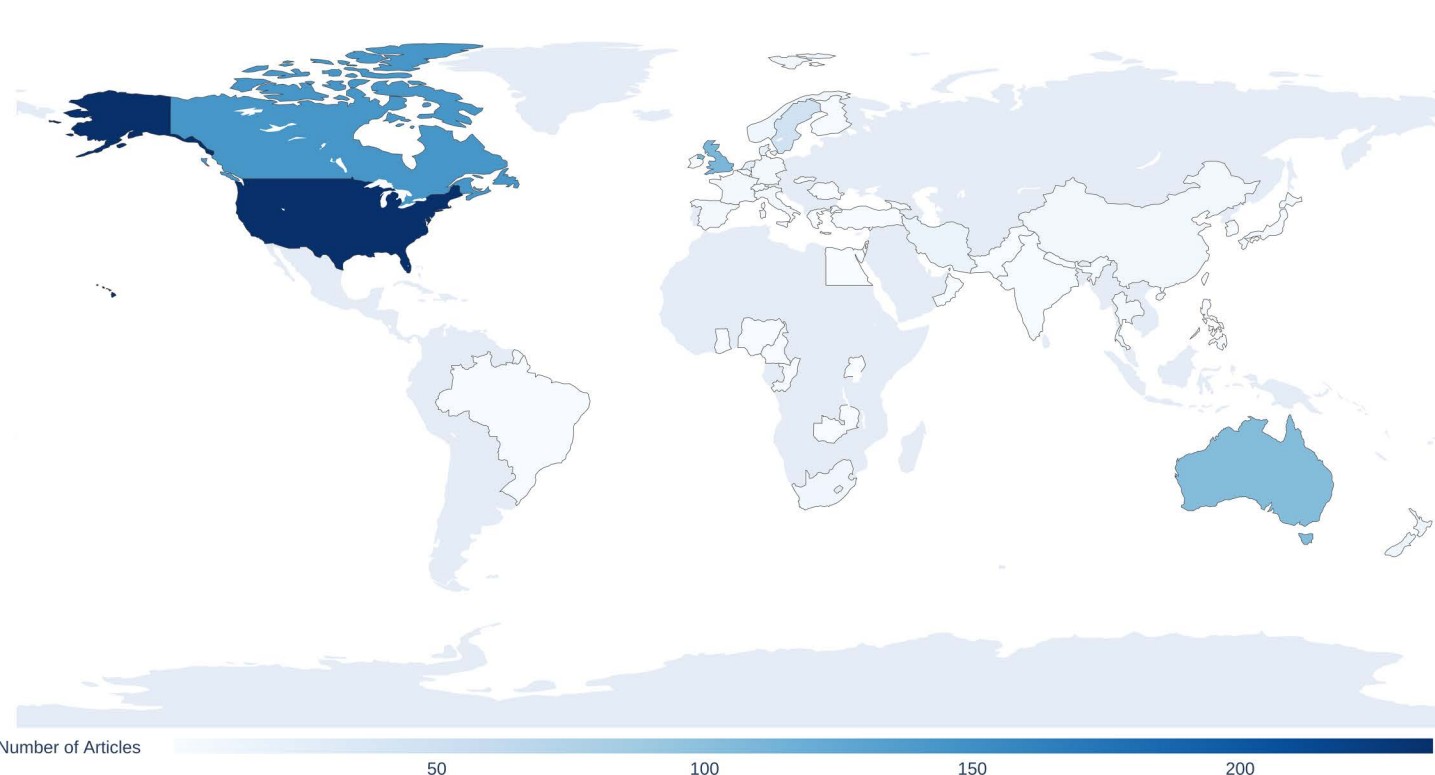

Articles by Country

Number of Articles

| 50 | 100 | 150 | 200 |

**Fig 5. World choropleth.** World map showing the number of publications by lead author country of origin (Geographic visualisation of citation counts per country was performed using GeoPandas [40]).

photovoice (n = 1), drawing methods (n = 3) and participatory diagramming (n = 1). Single methods were used in n = 661 (81.4%) in contrast to n = 151 (18.6%) using multiple methods.

Aggregating the analytical approach reported in papers was difficult due to the lack of consistency and heterogeneity of analytical approaches used despite many being variants of the same base approach, e.g., a variant of content analysis or a grounded theory style of analysis. Therefore, to understand the preferred analytical approaches used in the studies contents of this field were added to a word cloud generator. Three high word frequencies were identifiable: 'not reported'; Braun and Clarke (associated with thematic and reflexive thematic analysis) [41,42]; and Strauss and Corbin (associated with grounded theory informed analysis, e.g., constant comparison and open and axial coding) [43,44]. 'Not reported' was reported in n = 131 (16.1%) papers and referred to papers that provided no theoretical foundation for their analysis.

Information pertaining to the use of reporting checklists found that only n = 111 (13.7%) explicitly stated in their manuscript that they had used one, with the Consolidated Criteria for Reporting Qualitative Research (COREQ) [45] (n = 79, 9.7%) being the most common.

### The 'where'

The fourth theme examined where qualitative research was being published, and the publishing trends over time. Most commonly qualitative research is published in non-neurosurgical specific journals including: Brain Injury (n = 59); Disability and Rehabilitation (n = 57), and Neuropsychological Rehabilitation (n = 34). Only 8.4% (n = 68) were published in a

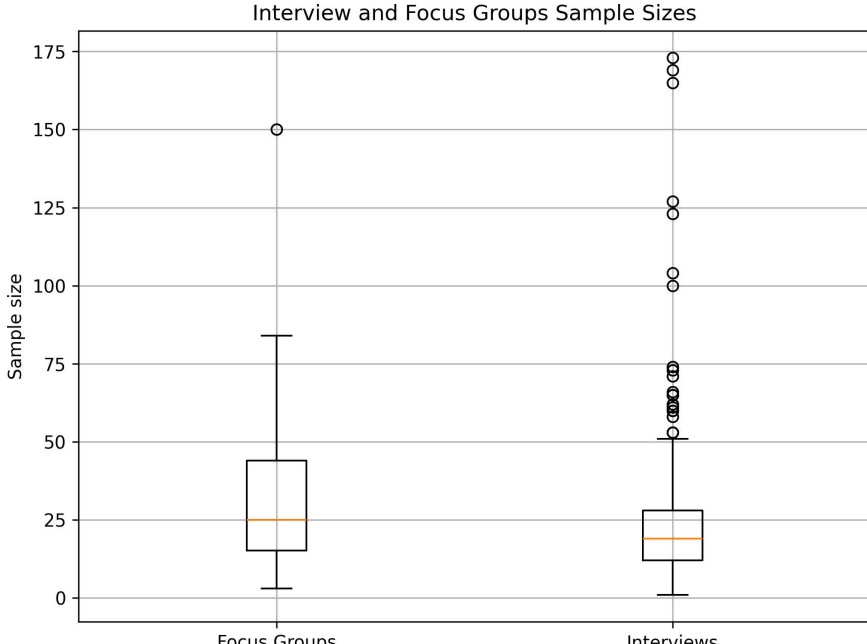

**Fig 6. Sample sizes.** Box plot of sample size variability for studies using interview and focus group methods.

neurosurgical specific journal: World Neurosurgery (n = 20), British Journal of Neurosurgery (n = 10), Acta Neurochirurgica (n = 10), and Journal of Neurosurgery (n = 10). An increasing trend for the publication of qualitative research over time was identified (see Fig 7).

### The 'use'

The final theme considers how qualitative research is being used; however, only 752 outputs had digital object identifiers (DoIs) from 1996 onwards amendable to bibliometric analysis. In this reduced data set the average FWCI was 0.96 peaking in 2006 1.78 and relatively consistent since 2017 (see Fig 8). Similarly, the FWVI was 1.11 and has also remained consistent since 2017. Analysis of policy impact found 18.9% (n = 142) of 752 scholarly outputs were cited in policies in 15 different countries. The four top performing outputs in terms of policy citation were: Perceived service and support needs during transition from hospital to home following acquired brain injury [46]; A qualitative study of the transition from hospital to home for individuals with acquired brain injury and their family caregivers [47]; Caring for the brain tumor patient: Family caregiver burden and unmet needs [48]; and The 'window of opportunity' for death after severe brain injury: Family experiences [49].

### Discussion

Our analysis shows the wide range of qualitative methods used to explore the subjective experience of patients and families living with neurosurgical conditions and the neurosurgeons who treat them. Although the evidence for this scoping review appears substantial, only 8.4% is published in neurosurgical specific journals, mirroring that of qualitative research in general surgery [14], suggesting qualitative methods are still underrepresented. Camlin and Seely [9] describe their experience of conducting qualitative research alongside large HIV research trials and noted their qualitative findings were not accepted into the same journal as the main trial paper. They specifically advocate for fully integrated manuscripts or two papers 'of equal weight' in the same journal issue so that important insight into trial findings are not lost. An apparent

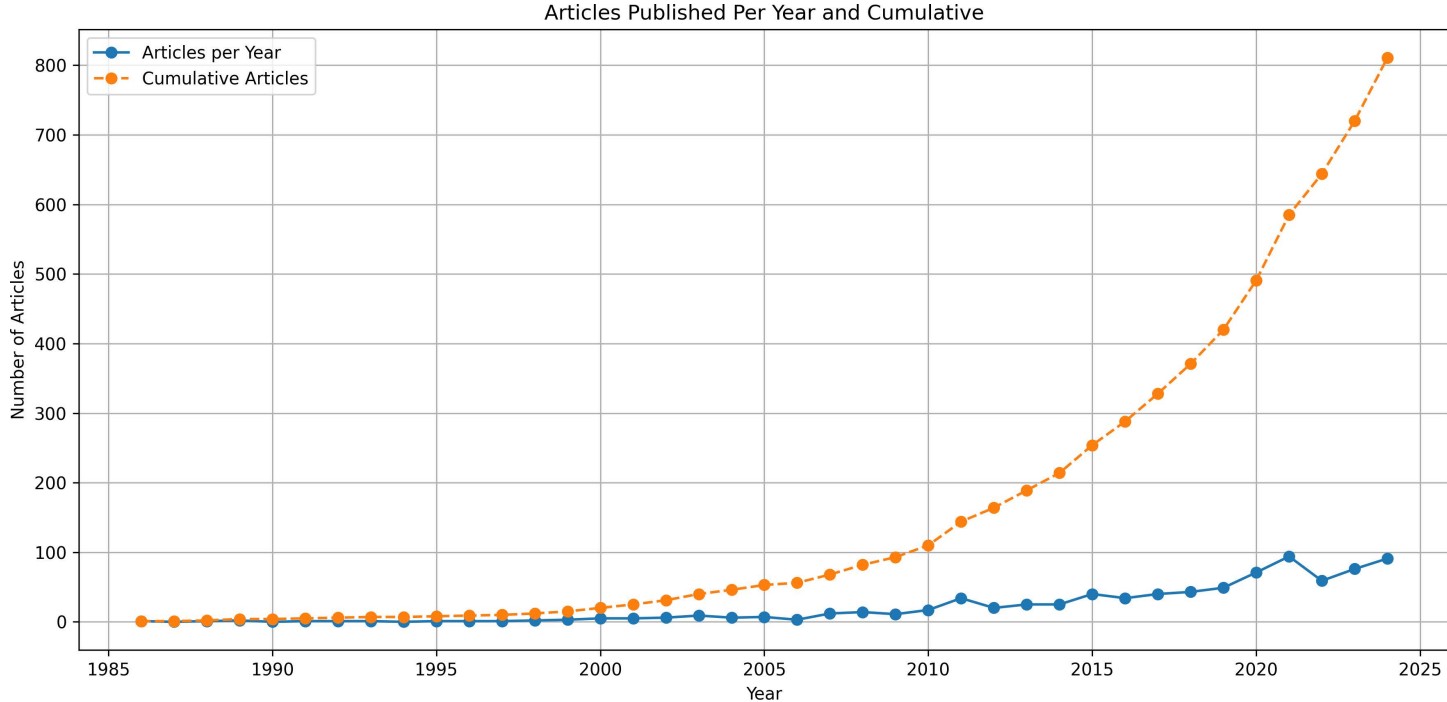

**Fig 7. Articles over time.** Line graph of publications over time.

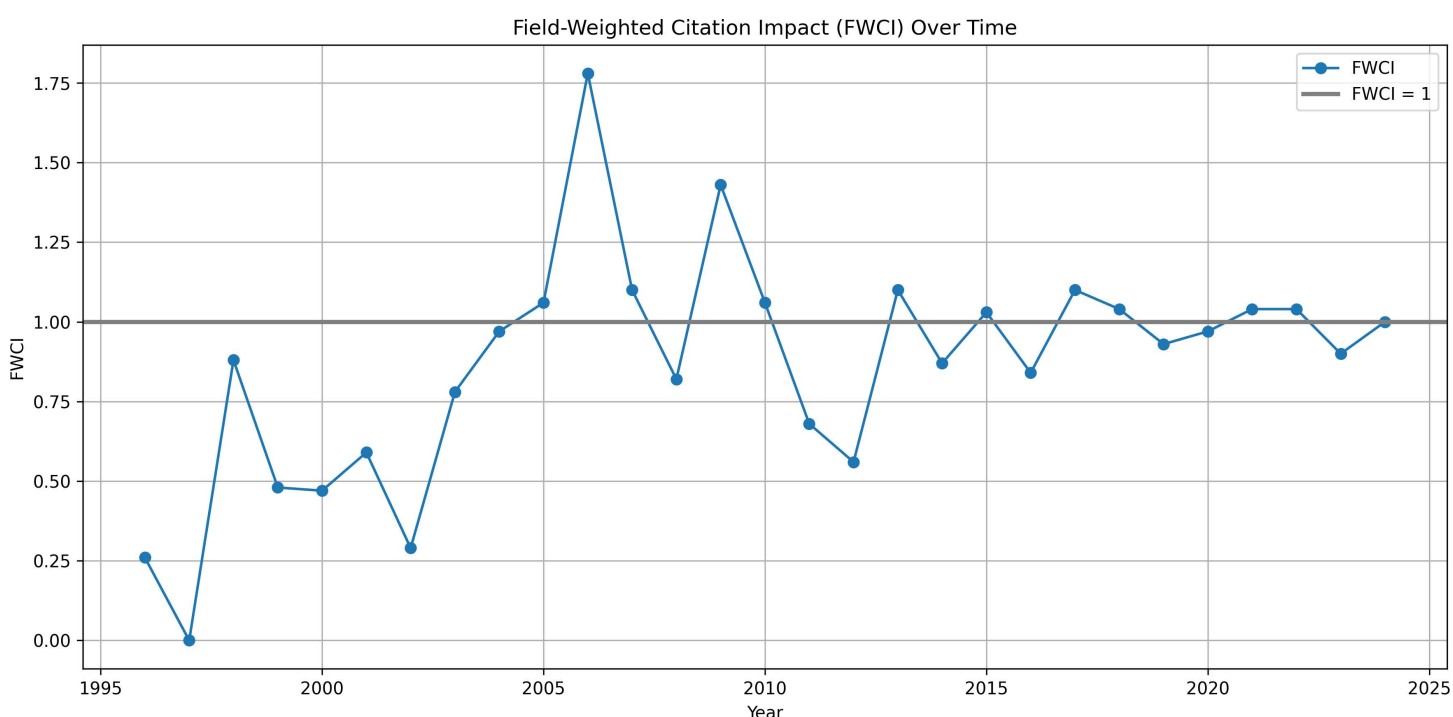

**Fig 8. Field weight citation impact.** Line graph of FWCI over time with a value of 1 highlighted representing performing as expected.

misunderstanding about quality has been cited as one reason for the lack of qualitative research in the medical literature while others suggest competing values within medical research may be responsible [50]. The Lancet Neurology Commission on TBI repeated calls for more recognition by peers and policy makers of the value of qualitative research [1,10]. This starts by increasing its visibility in influential journals. Editorial policies themselves do not often explicitly exclude qualitative research; therefore, we recommend future research about the barriers and facilitators of conducting and publishing high-quality qualitative research in neurosurgery.

Through this scoping review we have been able to take a wide-angle view at the qualitative neurosurgical evidence base and explore its use. Good qualitative research shifts the focus of inquiry from an assumed, often deductive form of knowledge (i.e., hypothesis-led), to a humbler position of not knowing and a commitment to finding out. Using qualitative methods to explore patient decision making for example can reveal important contextual information about what influences choice (see 'factors that influence the vestibular schwannoma treatment decision' [51] and 'Surgical decision-making among patients with uncontrolled epilepsy' [52]) or the acceptability of novel interventions and trial participation (see 'The Role of Family Members in Psychiatric Deep Brain Stimulation Trials' [53] 'Participant perceptions of changes in psychosocial domains following participation in an adaptive deep brain stimulation trial' [54], 'Posterior cervical foraminotomy versus anterior cervical discectomy for Cervical Brachialgia' [55].

Understanding the decision making process of neurosurgeons themselves positions neurosurgery as an art as well as a science (see 'Factors That Influence Intraoperative Decision-Making among Pediatric Neurosurgeons' [56] and 'Decision-making about intracranial pressure monitor placement in children with traumatic brain injury' [57]). Qualitative methods can illuminate the similarities and differences between patients and providers (see 'What Families Need and Physicians Deliver' [58]). Qualitative research also has the power to draw the reader into the insider view '"It was five years of hell": Parental experiences of navigating and processing the slow and arduous time to pediatric resective epilepsy surgery' [59]and 'I miss being me: Phenomenological Effects of Deep Brain Stimulation' [60].

Global neurosurgery is another area where qualitative research offers a unique opportunity to understand the cultural complexities of neurosurgical care. Examples included the exploration of neurosurgical equipment donation from the UK and Ireland to LMICs within the African continent, [61] reasons for delayed spinal cord decompression in Iran, [62] research capacity of neurosurgeons in LMICs, [31] access to healthcare for children with neural tube defects in Zambia, [63] African families' perceptions of traumatic brain injury [64], attitudes to neurosurgery in a low-income country, [23] and a mixed methods approach to understanding the TBI pathway in Myanmar [65].

Little evidence of neo-colonial 'helicopter' research was identified where HIC teams were answering LMIC research questions without the input of LMIC collaborators. However, there were several studies led by HIC first authors with LMIC participants (our own included [11,31]). Also noted were papers where both lead and senior authors were from HICs [23,65–67] and the LMIC authors occupied what may be perceived as "less prestigious authorship positions" [31] (p.9). There may be contextual information that meant this was appropriate for these studies. However, LMIC authors are rarely first or last author and this under-representation can limit research capacity building in these countries [68]. This finding, together with the low publication of qualitative studies from LMICs, presents an opportunity for more research to be led by researchers in LMICs about their unique contextual issues in delivering global neurosurgical care.

In this scoping review it was not possible to complete a detailed quality assessment of the papers included. However, the reporting of author positionality, paradigms or theoretical frameworks, clarity in data analysis methods and the use of reporting checklists may be useful indicators of quality in this evidence base. We were unable to determine author background in 57.5% of studies, only 29.4% described a philosophical foundation or theoretical framework underpinning their research. Under data analysis 16.1% lacked any published analytical method and we found reference to modified techniques, methods that were "roughly followed" and combined analytical techniques without transparency of the explicit analytical steps undertaken by the researchers themselves. Finally, only 13.7% reported explicitly the use a checklist for

PLOS One | https://doi.org/10.1371/journal.pone.0330770   August 21, 2025                                    12 / 18

reporting qualitative research. This suggests that, while valuable research is being conducted, there remains a need to improve methodological transparency and reporting in this evidence base.

However, more relevant to qualitative research, than simply reporting professional background, would be the presence of deeper insights, captured through reflexivity statements, about the researcher's values (axiology) and how they positioned themselves in the study (positionality) (see Raffaele and Tinofirei [69], Brunsden et al. [70], Brackenridge et al. [71], Apantaku et al. [72] and Elliott et al. [73] as good examples). Positionality also extends to the positioning of the study itself and whether it is inductive, deductive, interpretive, descriptive, constructivist, pragmatist. In mixed methods studies the way in which the qualitative and quantitative data are created, analysed and triangulated should also be explained, e.g., if there is a dominant paradigm (see Liang at al. [74], Douglas [75] and O'Reilly et al. [76]).

In this evidence base the reporting of paradigms, philosophy and theoretical frameworks were inconsistent at best or entirely absent at worst. This lack of thought about the construction of knowledge from within a qualitative paradigm may reflect a lack of qualitative thinking by the authors and produce studies that are underdeveloped in their ability to reach rich in-depth understanding of the problem under investigation.

Studies we reviewed often used less complex research designs. Single methods done well, can reach sophisticated in-depth and nuanced understanding of complex phenomena. However, single cross-sectional methods, or short interviews with a long semi-structured schedule can limit the data and the insight gained. With 18.6% of the evidence base using plural methods there is scope to increase the complexity of methodological approaches used in the evidence base.

Using theory, generating theory, adopting an interpretive stance, and being more creative with the final findings can facilitate a deeper understanding of phenomena under investigation. Of course, less complex findings may accompany less complex research questions, but when findings are more nuanced and compelling, they can lead to a more in-depth understanding of the phenomenon under investigation. More use of figures, illustrations and analytical maps, to convey how themes, or equivalent, relate to each other or/and the phenomena under investigation are also encouraged (see Brunner [77], Downing [78], Whiffin [31] for examples).

Furthermore, we recommend more detailed reporting of analytical techniques. It is not enough to simply state 'we used a thematic analysis', 'we coded to identify themes' or 'we performed a qualitative analysis'. A clear audit trail is the hallmark of rigorous inquiry and should not be omitted from publications (see Cregan [79], Brassel [80] and Cabrera [81] as good examples).

Using checklists can facilitate greater transparency of the methods employed. While the COREQ checklist is not without its limitations, [82] such tools, when used well, can enhance reporting and this would have been advantageous to many of the studies in this scoping review. However, more inclusion of explicit statements on trustworthiness and rigour would be more relevant within qualitative research to include greater consideration of quality markers such as those described by Tracy [83]. Those wanting to learn about how to conduct and report qualitative research may find it helpful to access neuroqual.org an open-source qualitative resources for neurosurgeons and neurosurgery.

Despite these challenges the analysis showed that as a publication set these outputs were performing almost as expected against similar publications from the same field in terms of citation and just above for views. Citation of these outputs in policy documents worldwide demonstrates the reach and influence of qualitative methodologies in shaping neurosurgical practice including guidelines for chronic pain, head injury, paediatric cancer, epilepsy and sub-arachnoid haemorrhage [84–88] amongst others. These examples highlight the opportunity to further integrate qualitative approaches into neurosurgical research, particularly in relation to reducing the prevalence of neurosurgical conditions, improving outcomes for neurosurgical patients and family, and advancing neurosurgical practice.

### Strengths and limitations

To some extent the results of this review were expected, i.e., that few studies are led by neurosurgeons or published in neurosurgical journals. However, the value of this review is the evidence that qualitative research is influencing

neurosurgical policy and practice. The review also highlights important gaps in the conduct and reporting of qualitative methods and provides useful guidance on how to grow the qualitative evidence base in the future and thus its impact. While the main aim of a scoping review, to determine the nature and volume of an evidence base, was achieved, further reviews or qualitative evidence syntheses on sub-sets of the evidence base with closer scrutiny of the findings and contribution to practice is recommended to understand how this evidence base to advancing the neurosurgical field.

The decision to exclude interventions that were more aligned to neurorehabilitation than neurosurgery was based on the need to draw a clear boundary around the phenomenon of interest; however, the subjective interpretation of this criterion is acknowledged.

Data extraction was based on the published manuscript despite additional details reported in online sources, supplementary files (unless explicitly signposted from the manuscript) or personal knowledge. This made extraction more reliable but may have missed important details. Affiliations of the wider author teams also went unanalysed which may have revealed a larger number of studies co-authored by neurosurgeons or affiliated with neurosurgical centres/departments. Furthermore, the bibliometric analysis identified several high-performing studies, unfortunately we were unable to assess their specific contribution to policy due to limitations in availability and the language of source documents. This means that certain findings will be under/over reported.

Finally, while it is reassuring to see an increase in qualitative studies in the evidence base, the quality of these studies was not assessed. There are significant areas in need of improvement, those conducting and reviewing qualitative research are urged to pay attention to methodological rigour to ensure both the quantity and quality of the evidence base increases over time.

## Conclusions

This scoping review is the first to our knowledge to comprehensively describe the range and reach of qualitative research in neurosurgery from the perspective of neurosurgeons, patients and families. Qualitative research is increasingly present in the evidence base, influencing neurosurgical practice and yet still underrepresented in neurosurgical journals. If good clinical practice relies on knowledge generated from in-depth qualitative studies as well as knowledge generated through large clinical trials the low rate of qualitative research in the neurosurgical evidence base may leave important questions unanswered. However, addressing the limitations of the evidence base is important to improve the quality, depth and contribution of this methodology to advancing practice.

## Supporting information

**S1 Table. PRISMA-ScR completed checklist.** Preferred Reporting Items for Systematic reviews and Meta-Analyses extension for Scoping Reviews.
(DOCX)

**S2 Table. Scoping review search strategy.** Full electronic search strategy deployed in Medline.
(DOCX)

**S3 Table. Data extraction.** Data extraction for papers included in this scoping review.
(PDF)

## Acknowledgments

We would like thank Emma Butler, Research & Publication Practice Manager and Naomi Richards, Research Intelligence Librarian at the University of Derby for their support with bibliometric analysis and interpretation.

## Author contributions

**Conceptualization:** Charlotte Jane Whiffin, Kathleen Joy O. Khu,, Brandon G. Smith.

**Data curation:** Brandon G. Smith.

**Formal analysis:** Charlotte Jane Whiffin, Kathleen Joy O. Khu,, Brandon G. Smith, Santhani M. Selveindran, Laura Hobbs, Samin Davoody, Yusuf Docrat, Orla Mantle, Upamanyu Nath, Lara Onbasi, Stasa Tumpa.

**Funding acquisition:** Angelos G. Kolias, Peter J. Hutchinson.

**Investigation:** Charlotte Jane Whiffin.

**Methodology:** Charlotte Jane Whiffin, Kathleen Joy O. Khu,, Isla Kuhn, Santhani M. Selveindran.

**Project administration:** Charlotte Jane Whiffin.

**Resources:** Brandon G. Smith.

**Software:** Isla Kuhn.

**Supervision:** Angelos G. Kolias, Peter J. Hutchinson.

**Writing – original draft:** Charlotte Jane Whiffin, Brandon G. Smith.

**Writing – review & editing:** Kathleen Joy O. Khu,, Brandon G. Smith, Laura Hobbs, Samin Davoody, Yusuf Docrat, Orla Mantle, Upamanyu Nath, Lara Onbasi, Stasa Tumpa, Ignatius N. Esene, Harry Mee, Fergus Gracey, Shobhana Nagraj, Tom Bashford, Angelos G. Kolias, Peter J. Hutchinson.

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
