## [Decision Letter · Decision Letter 0]

5 Jun 2025

Dear Dr. Whiffin,

Thank you for submitting your manuscript to PLOS ONE. After careful consideration, we feel that it has merit but does not fully meet PLOS ONE’s publication criteria as it currently stands. Therefore, we invite you to submit a revised version of the manuscript that addresses the points raised during the review process.

We look forward to receiving your revised manuscript.

With best wishes,

Dr. Mohammad Mofatteh, PhD, MPH, MSc, PGCert, BSc (Hons), MB BCh BAO (c)

Academic Editor

PLOS ONE

https://www.linkedin.com/in/drmofatteh/

Journal Requirements:

This research was funded by the NIHR (NIHR132455) using UK international development funding from the UK Government to support global health research. The views expressed in this publication are those of the author(s) and not necessarily those of the NIHR or the UK government.

3. Please amend your authorship list in your manuscript file to include author Charlotte Jane Jane Whiffin.

4. Please amend the manuscript submission data (via Edit Submission) to include author Charlotte J Whiffin.

5. We note that Figure 4 in your submission contain [map/satellite] images which may be copyrighted. All PLOS content is published under the Creative Commons Attribution License (CC BY 4.0), which means that the manuscript, images, and Supporting Information files will be freely available online, and any third party is permitted to access, download, copy, distribute, and use these materials in any way, even commercially, with proper attribution. For these reasons, we cannot publish previously copyrighted maps or satellite images created using proprietary data, such as Google software (Google Maps, Street View, and Earth). For more information, see our copyright guidelines: http://journals.plos.org/plosone/s/licenses-and-copyright.

a. You may seek permission from the original copyright holder of Figure 4 to publish the content specifically under the CC BY 4.0 license.

6. Please remove all personal information, ensure that the data shared are in accordance with participant consent, and re-upload a fully anonymized data set.

Additional Editor Comments :

Thank you for submitting your manuscript. Please provide a point-by-point response to the reviewers' comments along with a revised version of the manuscript.

Reviewers' comments:

Reviewer's Responses to Questions

**Comments to the Author**

1. Is the manuscript technically sound, and do the data support the conclusions?

Reviewer #1: Yes

Reviewer #2: Yes

2. Has the statistical analysis been performed appropriately and rigorously?

Reviewer #1: Yes

Reviewer #2: Yes

3. Have the authors made all data underlying the findings in their manuscript fully available?

Reviewer #1: Yes

Reviewer #2: Yes

4. Is the manuscript presented in an intelligible fashion and written in standard English?

Reviewer #1: Yes

Reviewer #2: Yes

Reviewer #1: This is a comprehensive scoping review assessing qualitative research in neurosurgery. It addresses a gap long acknowledged in neurosurgical literature and aligns with recent calls for qualitative integration. The manuscript highlights disparities in authorship and data collection between HICs and LMICs and addresses concerns about "helicopter research" and representation. Almost 19% of the reviewed outputs influenced policy across fifteen countries - a compelling argument for qualitative research utility in neurosurgery.

My Comments:

1. Quality Assessment:

Although scoping reviews typically avoid formal critical appraisal, a brief evaluation of methodological quality (eg Frequency of use of COREQ, audit tails, rigor indicators, reported paradigms, neurosurgeon involvement) would add weight.

2. Visual data representation

Analytical maps or conceptual/flow diagrams linking themes would enhacce readability

3. Propose a framework for future research

Based on the findings, suggest a conceptual model or framework for how qualitative research should be intergrated into neurosurgery moving forward

4. Highlight clinical impact examples

Include a short boxed section or table showing a few high-impact studies (cited in policy, guideline-forming) and thier qualitative contribution to practice change

Overall, this is a strong manuscript that makes a critical and timely contribution to both neurosurgery and qualitative health research

Recommendation: Minor Revision

Reviewer #2: This is a well written paper that addresses a clear need in the neurosurgical literature. I commend the authors for taking on this work, in this under researched area. The international collaboration on this paper only adds to its value and should also be commended.

**Do you want your identity to be public for this peer review?** For information about this choice, including consent withdrawal, please see our Privacy Policy

Reviewer #1: **Yes: ** Prajwal Ghimire

Reviewer #2: **Yes: ** Sanjeeva Jeyaretna

---

## [Author Response · Author response to Decision Letter 1]

23 Jul 2025

Please find below a summary of revisions made in responds to each of the points raised.

1. Please ensure that your manuscript meets PLOS ONE's style requirements, including those for file naming

I have reviewed the formatting conventions and updated the manuscript. I have removed additional details not requested; however, retained the funding statement on the title page. I hope this is acceptable.

A number of revisions have been made throughout the manuscript to align with the formatting requirements on the ‘title_authors_affiliations’ and ‘main body’ templates.

2.Please state what role the funders took in the study

Thank you for confirming that you will change the online submission on our behalf. Please find the correct statement for inclusion.

‘The funders had no role in study design, data collection and analysis, decision to publish, or preparation of the manuscript’

3. Please amend your authorship list in your manuscript file to include author Charlotte Jane Jane Whiffin.

4. Please amend the manuscript submission data (via Edit Submission) to include author Charlotte J Whiffin.

My apologies the error has been corrected to reflect one author Charlotte J Whiffin

5. We note that Figure 4 in your submission contain [map/satellite] images which may be copyrighted

Fig 4. Is a choropleth generated using GeoPanda. We have now referenced this python tool for geographic data next to the figure citation (now Fig 5).

Line 233

Added - (Geographic visualisation of citation counts per country was performed using GeoPandas [40])

6. Please remove all personal information, ensure that the data shared are in accordance with participant consent, and re-upload a fully anonymized data set.

Please may I seek further clarification on this point. The data set uploaded is a data extraction file of published evidence. Participant consent is not relevant to this scoping review.

7. Please review your reference list to ensure that it is complete and correct. If you have cited papers that have been retracted, please include the rationale for doing so

The reference list has been checked and to the best of my knowledge is accurate. I am not aware of any papers that have been retracted.

Reviewer #1: This is a comprehensive scoping review assessing qualitative research in neurosurgery. It addresses a gap long acknowledged in neurosurgical literature and aligns with recent calls for qualitative integration. The manuscript highlights disparities in authorship and data collection between HICs and LMICs and addresses concerns about "helicopter research" and representation. Almost 19% of the reviewed outputs influenced policy across fifteen countries - a compelling argument for qualitative research utility in neurosurgery.

Thank you for these kind comments.

1. Quality Assessment:

Although scoping reviews typically avoid formal critical appraisal, a brief evaluation of methodological quality (eg Frequency of use of COREQ, audit tails, rigor indicators, reported paradigms, neurosurgeon involvement) would add weight.

Thank you for this suggestion. We have now summarised the data in the results section that are indicative of quality and referred directly to these in the discussion.

Line 369-380

2. Visual data representation

Analytical maps or conceptual/flow diagrams linking themes would enhance readability

Thank you for this helpful suggestion. As this review presents a descriptive rather than interpretive analysis, a conceptual or analytical framework was not developed. However, to enhance readability and visualise the thematic structure, we have included a sunburst diagram that illustrates the distribution of coding across themes and sub-themes within the evidence base.

Line 191 & 195

New Fig 2. Sunburst chart of themes and sub-themes

3. Propose a framework for future research

Based on the findings, suggest a conceptual model or framework for how qualitative research should be integrated into neurosurgery moving forward

Thank you for this thoughtful suggestion. While the primary aim of this review was to descriptively synthesise the current qualitative literature in neurosurgery, rather than develop a conceptual model, we recognise the value of guiding future research in this area. Therefore, we have added a brief section to the discussion highlighting opportunities for integrating qualitative approaches into key areas of neurosurgical research

Line 429-431

4. Highlight clinical impact examples

Include a short boxed section or table showing a few high-impact studies (cited in policy, guideline-forming) and their qualitative contribution to practice change

Thank you for this helpful suggestion. We explored this recommendation in depth, including consultation with experts in bibliometric analysis. In doing so, we identified several limitations with the available SciVal data, which is based on Overton policy citation metrics.

First, our original manuscript cited the top five papers in terms of policy citations. However, on closer inspection, we found that the paper ‘I miss being me’ by Gilbert et al. (2017) was cited in only two unique policy documents, not five as initially reported. This discrepancy arose because the Overton database attributed multiple citations to the same policy document hosted on different webpages of EU organisations, thereby inflating the policy citation count.

We have therefore revised the manuscript to cite only the top four performing papers as fifth position is shared by multiple manuscripts, each cited in four policy documents, making it difficult to objectively prioritise a single additional study.

Second, we attempted to construct a summary table showing the study, citing policy, country, and year of citation, along with a qualitative review of each study’s contribution to policy or practice. However, several of the policy documents were either no longer publicly available or not written in English. This prevented a consistent and reliable analysis of the qualitative contribution of these studies to policy or guideline development.

As a result, we have not included a summary table but now address the limitations of bibliometric analysis more explicitly in the manuscript's limitations section.

Line 299

Line 450 – 453

5. Overall, this is a strong manuscript that makes a critical and timely contribution to both neurosurgery and qualitative health research

Many thanks, we are pleased this manuscript was well-received.

# Reviewer 2

This is a well written paper that addresses a clear need in the neurosurgical literature. I commend the authors for taking on this work, in this under researched area. The international collaboration on this paper only adds to its value and should also be commended.

We are very grateful for these encouraging comments.

---

## [Editor Report · Decision Letter 1]

6 Aug 2025

The range and reach of qualitative research in neurosurgery: A scoping review.

PONE-D-25-20297R1

Dear Dr. Whiffin,

We’re pleased to inform you that your manuscript has been judged scientifically suitable for publication and will be formally accepted for publication once it meets all outstanding technical requirements.

Kind regards,

Dr. Mohammad Mofatteh, PhD, MPH, MSc, PGCert, BSc (Hons), MB BCh BAO (c)

Academic Editor

PLOS ONE

Additional Editor Comments (optional):

Congratulations.
---

## [Editor Report · Acceptance letter]

PONE-D-25-20297R1

PLOS ONE

Dear Dr. Whiffin,

I'm pleased to inform you that your manuscript has been deemed suitable for publication in PLOS ONE. Congratulations! Your manuscript is now being handed over to our production team.

Kind regards,

on behalf of

Dr. Mohammad Mofatteh

Academic Editor

PLOS ONE